# Rosmarinic Acid, a Bioactive Phenolic Compound, Inhibits Glutamate Release from Rat Cerebrocortical Synaptosomes through GABA_A_ Receptor Activation

**DOI:** 10.3390/biom11071029

**Published:** 2021-07-15

**Authors:** Che-Chuan Wang, Pei-Wen Hsieh, Jinn-Rung Kuo, Su-Jane Wang

**Affiliations:** 1Chi Mei Medical Center, Department of Neurosurgery, Tainan 71004, Taiwan; wangchechuan@gmail.com (C.-C.W.); koujinnrung@gmail.com (J.-R.K.); 2Biotechnology, Southern Taiwan University of Science and Technology, Tainan 71005, Taiwan; 3Research Center for Chinese Herbal Medicine, College of Human Ecology, Chang Gung University of Science and Technology, Taoyuan 33303, Taiwan; pewehs@mail.cgu.edu.tw; 4Graduate Institute of Natural Products, School of Traditional Chinese Medicine, and Graduate Institute of Biomedical Sciences, College of Medicine, Chang Gung University, Taoyuan 33303, Taiwan; 5Department of Anesthesiology, Chang Gung Memorial Hospital, Taoyuan 33303, Taiwan; 6School of Medicine, Fu Jen Catholic University, New Taipei City 24205, Taiwan

**Keywords:** rosmarinic acid, glutamate release, GABA_A_ receptor, voltage-gated Ca^2+^ channel, CaMKII, synapsin I, synaptosome

## Abstract

Rosmarinic acid, a major component of rosemary, is a polyphenolic compound with potential neuroprotective effects. Asreducing the synaptic release of glutamate is crucial to achieving neuroprotectant’s pharmacotherapeutic effects, the effect of rosmarinic acid on glutamate release was investigated in rat cerebrocortical nerve terminals (synaptosomes). Rosmarinic acid depressed the 4-aminopyridine (4-AP)-induced glutamate release in a concentration-dependent manner. The removal of extracellular calcium and the blockade of vesicular transporters prevented the inhibition of glutamate release by rosmarinic acid. Rosmarinic acid reduced 4-AP-induced intrasynaptosomal Ca^2+^ elevation. The inhibition of N-, P/Q-type Ca^2+^ channels and the calcium/calmodulin-dependent kinase II (CaMKII) prevented rosmarinic acid from having effects on glutamate release. Rosmarinic acid also reduced the 4-AP-induced activation of CaMKII and the subsequent phosphorylation of synapsin I, the main presynaptic target of CaMKII. In addition, immunocytochemistry confirmed the presence of GABA_A_ receptors. GABA_A_ receptor agonist and antagonist blocked the inhibitory effect of rosmarinic acid on 4-AP-evoked glutamate release. Docking data also revealed that rosmarinic acid formed a hydrogen bond with the amino acid residues of GABA_A_ receptor. These results suggested that rosmarinic acid activates GABA_A_ receptors in cerebrocortical synaptosomes to decrease Ca^2+^ influx and CaMKII/synapsin I pathway to inhibit the evoked glutamate release.

## 1. Introduction

Glutamate is the main excitatory neurotransmitter of the central nervous system (CNS) and acts through glutamate receptors to modulate synaptic transmission. Under physiological conditions, glutamatergic neurotransmission is required for normal neuronal function and is involved in learning and memory processes [1]. However, excessive glutamate release successively causes the overactivation of glutamate receptors, highlevels of Ca^2+^ entry into neurons, protease activation, mitochondrial damage, free radical generation, and finally neuronal death [2,3]. This process, known as excitotoxicity, is the mechanism underlying numerous CNS diseases [4,5]. Modulating synaptic glutamate release is thus a valuable strategy for reducing neurotoxicity and protecting the brain [6,7,8].

Plant-derived compounds have received considerable attention for their neuroprotective activities [9,10]. Rosemary (*Rosmarinus officinalis* L.), a common household plant, is a herb used in cooking, teas, and folk remedies [11]. In folk and traditional medicine, rosemary is used for the treatment of headache, dysmenorrhea, stomachache, insomnia, hysteria, epilepsy, and depression [12]. Its antiviral [13], antioxidant [14], anti-inflammatory [15], anti-tumorigenic [16], antinociceptive [17], antidepressant [18], neuroprotective [19], and memory-enhancing properties [20], among others, have been reported. We focused our bioactivity analysis on rosmarinic acid (Figure 1A), which has been indicated to possess neuroprotective effects in various models of ischemia and neurodegeneration [21,22], as well as in chemical-induced neurotoxicity and oxidative stress [23,24,25]. The neuroprotective properties of rosmarinic acid and the involvement of the glutamatergic system in excitotoxicity [2,4] prompted us to evaluate its potential role in the regulation of glutamate release. Therefore, this study used rat cerebrocortical nerve terminals (synaptosomes) to investigate the effect of rosmarinic acid on glutamate release and elucidate the relevant mechanisms.

## 2. Materials and Methods

### 2.1. Animals

Male Sprague–Dawley rats (150–200 g, BioLASCO Inc., Taipei, Taiwan) were cared for in accordance with the guidelines of the Care and Use of Laboratory Animals (NAC 2011). All the procedures were approved by the Fu Jen Institutional Animal Care and Utilization Committee (IACUC No. A10812).

### 2.2. Isolation of Synaptosomes from the Rat

Synaptosomes were isolated as previously described [26]. Animals were sacrificed via cervical dislocation and the cerebral cortex were rapidly removed. The brain tissue was homogenized in 320 mM sucrose solution and centrifuged at 3000× *g* for 10 min. The supernatant was stratified on Percoll discontinuous gradients and centrifuged at 32,500× *g* for 7 min. The synaptosomal fraction was collected and centrifuged for 10 min at 27,000× *g*. Protein concentration was determined using the Bradford assay. Synaptosomes were centrifuged at 3000× *g* for 10 min in the final wash to obtain synaptosomal pellets with 0.5 mg protein.

### 2.3. Measurement of Glutamate Release

For the glutamate release experiments, the synaptosomal pellet (0.5 mg protein) was resuspended in the Hepes-buffered solution and glutamate release was assayed by on-line fluorimetry [27]. CaCl_2_ (1.2 mM), glutamate dehydrogenase (GDH, 50 units/mL) and NADP^+^ (2 mM) were added at the start of incubation. Glutamate release was induced with 4-AP (1 mM) and monitored by measuring the increase of fluorescence (excitation and emission wavelengths of 340 and 460 nm, respectively) resulting from NADPH being produced by the oxidative deamination of released glutamate by GDH. Released glutamate was calibrated by a standard of exogenous glutamate (5 nmol) and expressed as nanomoles glutamate per milligram synaptosomal protein (nmol/mg).

### 2.4. Measurement of Intrasynaptosomal Ca^2+^ Concentration ([Ca^2+^]_i_)

Synaptosomes (0.5 mg protein) were incubated in the Hepes-buffered solution containing Fura 2-AM (5 μM) and CaCl_2_ (0.1 mM) for 30 min at 37 °C. Samples were centrifuged for 1 min at 5000 rpm, and pellets were resuspended in Hepes-buffered medium containing CaCl_2_ (1.2 mM). Fura-2-Ca fluorescence was monitored at 5 s intervals for 5 min. [Ca^2+^]_i_ (nM) was calculated by using calibration procedures and equations described previously [28].

### 2.5. Measurement of Membrane Potential

The synaptosomal membrane potential was assayed with a positively charged membrane potential-sensitive carbocyanine dye DiSC_3_(5) [29]. DiSC_3_(5) fluorescence was monitored at 2 s intervals and data are expressed in fluorescence units.

### 2.6. Western Blot

Synaptosomal lysates (20μg per lane) were separated on 10% sodium dodecyl sulfate-polyacrylamide gel electrophoresis (SDS-PAGE) and transferred onto nitrocellulose membranes, incubated with anti-CaMKII (1:10,000, Cell Signaling, Beverly, MA, USA), anti-p-CMKII (1:2000, Cell Signaling, Beverly, MA, USA), anti-synapsin I (1:30,000, Cell Signaling, Beverly, MA, USA), and anti-p-synapsin I (Serine 603) (1:2000, GeneTex, Irvine, CA, USA) antibodies for overnight at 4 °C. This was followed by washing with Tris-buffered saline containing 0.1% Tween 20 (TBST) and incubation with horseradish peroxidase-conjugated secondary antibodies (1:5000, Cell Signaling, Beverly, MA, USA) for 2 h at room temperature. Immunoreactive bands were visualized by chemiluminescence (GeneTex, Irvine, CA, USA). Quantification was obtained by scanning densitometry of five independent experiments, and analyzed with Image J software (version 1.43, Bethesda, MD, USA) [30].

### 2.7. Immunocytochemistry

The synaptosomes were attached to the polylysine-coated coverslips for 40 min at 4 °C, fixed with 4% paraformaldehyde in 0.1 M phosphate buffer (pH 7.4) for 30 min, permeabilized with 0.2% Triton X-100 for 60 min and incubated for 24 h with a mixture primary antibodies against vesicular transporter of glutamate type 1 (VGLUT1, 1:200; Abcam, Cambridge, UK) and GABA_A_ receptorα1 subunit(1:100; Abcam, Cambridge, UK) for 90 min at room temperature. After rinsing with blocking buffer, the synaptosomes were incubated with a mixture of goat anti-mouse DyLight 549-and goat anti-rabbit fluoresceinisothiocyanate (FITC)-conjugated secondary antibodies (1:200; Invitrogen, Carlsbad, CA, USA) for 1 h at room temperature. The synaptosomes were then washed three times with phosphate buffer and 0.1 M carbonate buffer (pH 9.2), and the coverslips were mounted with fluoromount (DAKO North America, Inc., Carpinteria, CA, USA). Images were acquired in an Image Xpress micro confocal (Molecular Devices, San Jose, CA, USA).

### 2.8. Molecular Docking Study

The molecular docking experiment was performed using the tools of CDOCKER in Discovery Studio 4.1 client. The molecular structure of GABA_A_ receptor protein (PDB ID 4COF) from the RCSB Protein Data Bank was prepared with the standard protocol using Discovery Studio 4.1 client. The optimized structure of rosmarinic acid was created using flexibly docked following the default setting in software, and then inserted into the probable binding poses of the active site in protein.

### 2.9. Statistical Analysis

Reported data were expressed as means ± standard error of the mean (SEM). To assess statistical differences between two groups, the unpaired (two-tailed) Student’s *t* test was used. Multiple comparisons were performed with the analysis of variance (ANOVA) followed by Tukey’s post hoc test. Differences were considered as statistically significant at *p* < 0.05.

### 2.10. Chemicals

Rosmarinic acid (purity > 98%, ChemFace, Wuhan, China), bafilomycin A1 (Tocris Bioscience, Bristol, United Kingdom), dantrolene (Tocris Bioscience), 7-chloro-5-(2-chlorophenyl)-1,5-dihydro-4,1-benzothiazepin-2(3H)-one (CGP37157, Tocris Bioscience), *N*-[2-(p-bromocinnamylamino)ethyl]-5-isoquinolinesulfonamide (H89, Tocris Bioscience), bisindolylmaleimide I (GF109203X, Tocris Bioscience), 1-[N,O-Bis(5-isoquinolinesulfonyl)-*N*-methyl-l-tyrosyl]-4-phenylpiperazine (KN62, Tocris Bioscience), 3,3,3-dipropylthiadicarbocyanine iodide [DiSC3(5)] (Thermo, Waltham, MA, USA), and fura-2-acetoxymethyl ester (Fura-2-AM, Thermo) were made up in 10 mM stock solution in DMSO. DL-threo-β-benzyloxyaspartate (DL-TBOA, Tocris Bioscienc), ω-conotoxin GVIA (ω-CgTX GVIA, Alomone Lab, Jerusalem, Israel), and ω-agatoxin IVA (ω-Aga IVA, Alomone Lab) were made up in 5 mM stock solution in water. 4-aminopyridine (4-AP, Sigma-Aldrich, St. Louis, MO, USA) were made up in 300 mM stock solution in water.

## 3. Results

### 3.1. Effect of Rosmarinic Acid on the 4-AP-Evoked Glutamate Release in Rat Cerebrocortical Synaptosomes

To investigate the effect of rosmarinic acid on glutamate release, synaptosomes isolated from rat cerebral cortex were stimulated by 1 mM 4-AP, which opens voltage-gated Ca^2+^ channels (VGCCs) and induces the release of glutamate [31]. As shown in Figure 1B, preincubation with rosmarinic acid (30 μM) for 10 min before 4-AP addition did not produce any significant effect on the basal release of glutamate, but markedly reduced the 4-AP-induced release of glutamate release in the presence of 1.2 mM CaCl_2_ (*n* = 5, *p <* 0.001 vs. control group). At concentrations of 1–50 μM, the effects of rosmarinic acid were concentration-dependent (Figure 1C) with an IC_50_ of 11 μM and maximum inhibition of 48.8 ± 2.5%. In addition, 4-AP-evoked glutamate release was reduced in Ca^2+^-free media (plus EGTA) (*p <* 0.001). In Ca^2+^-free conditions (plus EGTA), rosmarinic acid failed to reduce the release of glutamate evoked by 4-AP (*n* = 5, *p* = 0.9; Figure 1C). On the other hand, 4-AP-evoked glutamate release was increased in the presence of DL-TBOA, an inhibitor of the plasma membrane glutamate transporter, which blocks the Ca^2+^-independent nonvesicular efflux by transporter reversal (*n* = 5, *p <* 0.001 vs. control group). With DL-TBOA present, rosmarinic acid significantly inhibited the 4-AP-evoked glutamate release (*n* = 5, *p <* 0.001 vs. the DL-TBOA-treated group, Figure 1C). By contrast, bafilomycin A1, an inhibitor of vesicular glutamate transporters, reduced 4-AP-evoked glutamate release (*p <* 0.001 vs. control group). In the presence of bafilomycin A1, rosmarinic acid failed to produce significant inhibition (*p* = 0.98 vs. bafilomycin A1-treated group, Figure 1C).

### 3.2. Effect of Rosmarinic Acid on [Ca^2+^]_i_ and Membrane Potential

Table 1 shows that 4-AP (1 mM) elicited a rise in [Ca^2+^]_i_ and rosmarinic acid preincubation reduced the 4-AP-induced [Ca^2+^]_i_ increase by 35.9 ± 1.7% (*n* = 5, *p <* 0.001 vs. control group). Rosmarinic acid had no significant effect on the basal [Ca^2+^]_I_ (*p* = 0.96). In addition, 4-AP (1 mM) evoked DiSC3(5) fluorescence increase and this effect was not affected by rosmarinic acid preincubation (*n* = 5, *p* = 0.9 vs. control group, Table 1).

### 3.3. Effect of Rosmarinic Acid on Glutamate Release in the Presence of VGCC Blockers or Intracellular Ca^2+^ Release Inhibitors

Either VGCCs or intracellular Ca^2+^ stores are responsible for the release of glutamate evoked by depolarization [32,33]. As shown in Figure 2, 4-AP-evoked glutamate release was reduced by ω-CgTX GVIA (*p <* 0.001 vs. control group) and ω-AgTX IVA (*p <* 0.001 vs. control group), which selectively block N- and P/Q-type Ca^2+^ channels, respectively [32]. With ω-CgTX GVIA or ω-Aga IVA present, 4-AP-evoked glutamate release was further inhibited by rosmarinic acid (*n* = 5, *p <* 0.001 vs. ω-CgTX GVIA- or ω-Aga IVA-treated group). However, the inhibitory action of rosmarinic acid was abolished in the presence of ω-CgTX GVIA and ω-Aga IVA. The release measured in the presence of ω-CgTX GVIA and ω-Aga IVA and rosmarinic acid wassimilar to that obtained in the presence of ω-CgTX GVIA and ω-Aga IVA (*n* = 5, *p* = 0.98 vs.ω-CgTX GVIA and ω-Aga IVA-treated group). In addition, 4-AP-evoked glutamate release was reduced by dantrolene, an inhibitor of intracellular Ca^2+^ release from endoplasmic reticulum (*p* < 0.001 vs. control group), and CGP37157, an inhibitor of mitochondrial Na^+^/Ca^2+^ exchange (*p* < 0.001 vs. control group). With dantrolene or CGP37157 present, rosmarinic acid was able to reduce 4-AP-evoked glutamate release (*n* = 5, *p* < 0.001 vs. the dantrolene- or CGP37157-treated group, Figure 3). Furthermore, rosmarinic acid preincubation efficiently decreased 15 mM KCl-evoked glutamate release (*n* = 5, *p <* 0.001 vs. control group, Figure 2, inset), a process that involves Ca^2+^ influx primarily through VGCC opening [34].

### 3.4. Effect of Rosmarinic Acid on Glutamate Release in the Presence of Protein Kinase Inhibitors

Figure 3 shows that 4-AP-evoked glutamate release was reduced in the presence of H89, a protein kinase A (PKA) inhibitor, GF109203X, a PKC inhibitor, and KN63, a CaMKII inhibitor (*p <* 0.001 vs. control group). With H89 or GF109203X present, rosmarinic acid could still reduce the 4-AP-evoked glutamate release (*p <* 0.001 vs. H89- or GF109203X-treated group). By contrast, rosmarinic acid did not produce any significant inhibition of glutamate release in the presence of KN62 (*n* = 7, *p* = 0.97 vs. the KN62-treated group).

### 3.5. Effect of Rosmarinic Acid on the Phosphorylation of CaMKII and Synapsin I (Serine 603, a Substrate Site of CaMKII) in Synaptosomes

Figure 4 shows a statistically significant increase in the phosphorylation of CaMKII and synapsin I was observed in the 4-AP group (*p <* 0.001 vs. control group). After pretreatment with rosmarinic acid, no significant increase in CaMKII and synapsin I phosphorylation was observed after exposure to 1 mM 4-AP compared with the control group (*n* = 5, *p* = 0.9, Figure 4).

### 3.6. Effect of Rosmarinic Acid on Glutamate Release in the Presence of GABA_A_Receptor Agonist and Antagonist

The central effects of rosmarinic acid have been shown to be associated with the modulation of γ-aminobutyric acid type A (GABA_A_) receptors [35,36], which led us to ask the question of whether the effect of rosmarinic acid on glutamate release was mediated by GABA_A_ receptors. We checked the presence of GABA_A_ receptors in rat cerebrocortical synaptosomes by co-labeling with antibodies against the GABA_A_receptorα1 subunit, and the vesicular transporter of glutamate type 1 (VGLUT1), a glutamatergic terminal marker protein. As shown in Figure 5A, confocal microscopy showed that VGLUT1-positive glutamatergic particles (red) was immunopositive for the GABA_A_ receptor α1 subunit (green). About 68.2 ± 1.5% of the VGLUT1-positive particles were also positive for GABA_A_ receptorα1 subunit (yellow). In addition, we examined the effect of isoguvacine, a GABA_A_ receptor agonist, and SR95531, a GABA_A_ receptor antagonist, on the action of rosmarinic acid. Figure 5B shows that the application of isoguvacine significantly reduced 4-AP-evoked glutamate release (*n* = 5, *p <* 0.001 vs. control group). With isoguvacine present, rosmarinic acid had no significant effect on 4-AP-evoked glutamate release (*p* = 0.9 vs. isoguvacine-treated group). A similar result was also obtained in the presence of SR95531. As shown in Figure 5B, rosmarinic acid failed to reduce 4-AP-evoked glutamate release in the presence of SR95531 (*n* = 5, *p* = 0.8 vs. SR95531-treated group). However, SR95531 had no significant effect on control 4-AP-evoked glutamate release (*p* = 0.2 vs. control group). Furthermore, to predict the binding mode of rosmarinic acid, the molecular docking within the active site of GABA_A_ receptor was based on the molecular structure (PDB ID 4COF) downloaded from RCSB protein data bank. The results indicated that rosmarinic acid formed six hydrogen-bonding interactions with the amino acid residues Asn149, Glu153, Arg192, Arg196, Ser209, and Ser211, respectively (Figure 6).

## 4. Discussion

This is the first study of the effects of rosmarinic acid on central glutamatergic transmission. This work was centered on the modulation by rosmarinic acid of glutamate release from rat cerebrocortical synaptosomes. Rosmarinic acid, through GABA_A_ receptor activation and CaMKII suppression, reduced 4-AP-evoked glutamate release.

Neuronal glutamate release is induced by 4-AP through two mechanisms: Ca^2+^-dependent vesicular release and Ca^2+^-independent release through the reversal of the plasma membrane glutamate uptake carrier [31,37]. In the current study, glutamate release inhibition by rosmarinic acid was prevented when external Ca^2+^ was removed, indicating the dependence of this inhibition on extracellular Ca^2+^. In addition, the blockage of vesicular release by bafilomycin A1, but not the blockage of glutamate transporters working in the reverse mode with DL-TBOA, abolished the rosmarinic acid-induced inhibition of glutamate release evoked by 4-AP. Our results indicate that the rosmarinic acid-mediated inhibition of glutamate release is mainly ascribable to a reduction in Ca^2+^-dependent exocytosis.

The modulation of 4-AP-evoked Ca^2+^-dependent glutamate release can occur at several putative sites in nerve terminals, including the Na^+^, K^+^, and Ca^2+^ channels. Modulation can also be achieved in the process of the release process itself. For instance, the inhibition of Na^+^ channels or the activation of K^+^ channels results in nerve terminal hyperpolarization, which causes a subsequent reduction in the presynaptic voltage-dependent neuronal entry of Ca^2+^ and a consequent reduction in glutamate release [38]. The present findings suggest that rosmarinic acid does not reduce synaptosomal excitability, which would in turn reduce the influx of Ca^2+^ and thereby reduce glutamate release. This is because rosmarinic acid significantly inhibited 4-AP- and KCl-induced glutamate release. Another reason is that glutamate release induced by 1mM 4-AP involved both Na^+^ and Ca^2+^ channels, whereas that induced by 15mM external KCl involved only Ca^2+^ channels [31]. In other words, Na^+^ channels were not involved in the inhibition of glutamate release by rosmarinic acid. Furthermore, no substantial effect of rosmarinic acid on synaptosomal membrane potential was observed either in the resting condition or during depolarization with 4-AP, thus indicating no effect on K^+^ conductance.

In addition, the inhibition of glutamate release by rosmarinic acid was associated with a reduction in 4-AP-evoked increase in [Ca^2+^]_i_. During the depolarization of synaptosomes in Ca^2+^-containing medium, the elevation of [Ca^2+^]_i_ is mediated by Ca^2+^ influx through VGCCs and Ca^2+^ release from internal stores [32,33,39]. The rosmarinic acid-induced inhibition of 4-AP-evoked glutamate release from depolarized synaptosomes was completely abolished when N- and P/Q-type Ca^2+^ channels were blocked. However, neither dantrolene, an inhibitor of the release of intracellular Ca^2+^ from the endoplasmic reticulum, nor CGP37157, a mitochondrial Na^+^/Ca^2+^ exchange blocker, affected the inhibitory effect of rosmarinic acid on 4-AP-evoked glutamate release. These results suggest that rosmarinic acid can inhibit glutamate release by reducing the activity of VGCCs. However, how rosmarinic acid affects VGCCs remains to be elucidated.

In synaptic terminals, CaMKII, a serine/threonine kinase activated by Ca^2+^ and calmodulin regulates neurotransmitter release [40]. CaMKII has been reported to increase glutamate release by phosphorylating numerous synaptic proteins, including synapsin I [41,42,43]. The phosphorylation of synapsin I causes synaptic vesicles to dissociate from the cytoskeleton, thereby increasing the proportion of nerve terminal vesicles that are available for release [44,45,46]. In our study, the inhibitory effect of rosmarinic acid on glutamate release was abolished by the CaMKII inhibitor KN62. Moreover, rosmarinic acid inhibited the 4-AP-evoked phosphorylation of CaMKII and synapsin I. These results suggest that the suppression of CaMKII activity and the inhibition of synapsin I phosphorylation is involved in the rosmarinic acid-mediated inhibition of glutamate release. In addition to synapsin I, however, several other presynaptic proteins, including synaptophysin, syntaxin, and SNAP 25, have been demonstrated to be phosphorylated by CaMKII [47,48]. The possibility that these proteins are involved in phosphorylation inhibition with regard to the inhibition of rosmarinic acid release cannot be excluded.

Rosmarinic acid is known to modulate the function of GABA_A_ receptors [35,36]. Therefore, the question arises of whether the rosmarinic acid effect observed in our study involves an action on GABA_A_ receptors. GABA_A_ receptors, a large and diverse family of Cl^−^-permeable ion channels, mediate inhibitory neurotransmission in the CNS [49]. GABA_A_ receptors are assembled from seven class of homologous subunits, α, β, γ, δ, ε, π, and θ, into heteropentamers. The majority of GABA_A_ receptors are composed of two α1, two β2, and one γ2 subunit (α1β2γ2 receptors), which are primarily located synaptically and are involved in the hypnotic/sedative effects [50,51]. In the present study, immunocytochemical analysis of rat synaptosomes demonstrated that the GABA_A_ receptor α1 subunitswere present on glutamatergic terminals, which is consistent with previous studies [52,53]. In addition, isoguvacine and SR95531, GABA_A_ receptor agonist and antagonist, respectively, blocked the inhibitory effect of rosmarinic acid on 4-AP-evoked glutamate release. Furthermore, our docking data revealed that rosmarinic acid can form hydrogen bond with the amino acid residues (Asn149, Glu153, Arg192, Arg196, Ser209, and Ser211) of GABA_A_ receptor. Since the amino acid residues Ser204, Tyr205, Arg207, and Ser209 in the loop C region of GABA_A_ protein are the key residues face into GABA binding, and the loop F region (Asp192-Arg197) could interact with neighboring residues by hydrogen bonding and hydrophobic interaction, which change conformation during receptor activation [54,55]. Rosmarinic acid may be able to modulate the GABA_A_ receptor activation through the interaction betweenArg192, Arg196, and Ser209. At the glutamatergic terminals, the activation of GABA_A_ receptors cause the hyperpolarization through increasing Cl^−^ influx, which would reduce action potentials. This reduction would decrease the presynaptic Ca^2+^ influx, which in turn would affect glutamate release [56]. In the present study, we did not examine the effect of rosmarinic acid on intracellular Cl^−^ influx in cerebrocortical synaptosomes. However, rosmarinic acid has been shown to increase intracellular Cl^−^ influx in primary cultured hypothalamic cells [35]. Based on these considerations, we infer that GABA_A_ receptor activation of glutamatergic terminals by rosmarinic acid results in hyperpolarization, which reduces Ca^2+^ influx and the amount of glutamate released. In fact, enhancement of GABA_A_ receptor activity has been championed by many studies as the potential target site for natural products [57,58].

## 5. Conclusions

In conclusion, our results demonstrate for the first time that rosmarinic acid, through GABA_A_ receptor activation to affect a reduction of Ca^2+^ influx and CaMKII/synapsin I pathway, inhibits glutamate release from rat cerebrocortical nerve terminals (Figure 7). As for glutamate, the neuroprotective properties of reduced glutamatergic neurotransmission are long-established [6,7,8]; therefore, the inhibition of glutamate release by rosmarinic acid may constitute an explanatory mechanism of its neuroprotective effects [21,22]. Although the relevance of our findings to clinical situations remains to be determined, they suggest that rosmarinic acid is a promising anti-excitotoxic drug for the treatment of neurological disorders involving glutamate excitotoxicity.

## Figures and Tables

**Figure 1 biomolecules-11-01029-f001:**
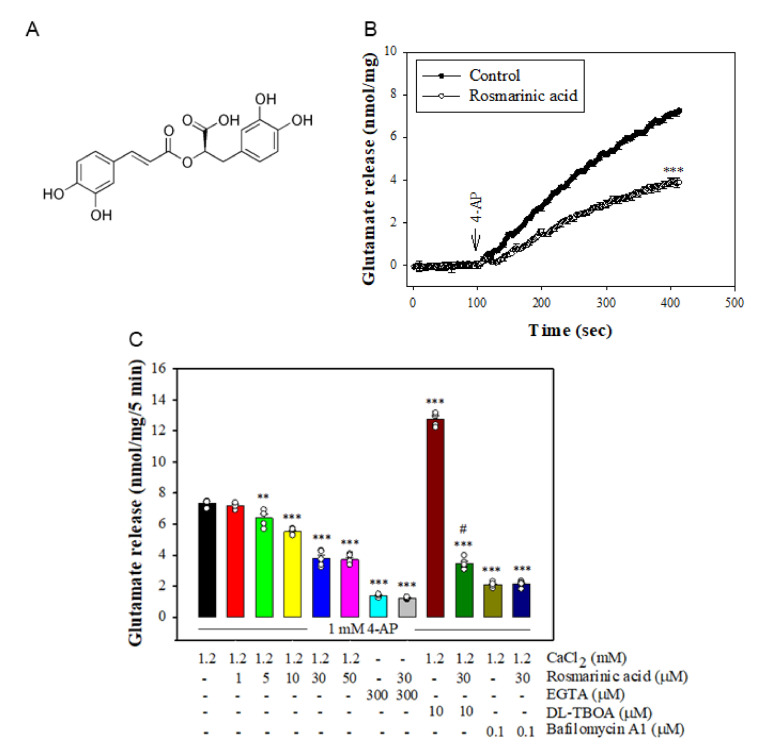
Effect of rosmarinic acid on 4-AP-evoked glutamate release from rat cerebrocortical nerve terminals. (**A**) The chemical structure of rosmarinic acid. (**B**) Glutamate release was measured under control conditions or in the presence of rosmarinic acid added 10 min prior to the addition of 4-AP. (**C**) Effect of rosmarinic acid at different concentrations on 4-AP-evoked glutamate release and extracellular Ca^2+^-free solution, glutamate transporter inhibitor DL-TBOA or vesicular release inhibitor bafilomycin A1 on the effect of rosmarinic acid. Rosmarinic acid, DL-TBOA or bafilomycin A1 was added 10 min before depolarization. Data are mean ± SEM (*n* = 5 per group). ** *p* < 0.01, *** *p* < 0.001 compared with the control group. ^#^
*p* < 0.001 compared with the DL-TBOA-treated group.

**Figure 2 biomolecules-11-01029-f002:**
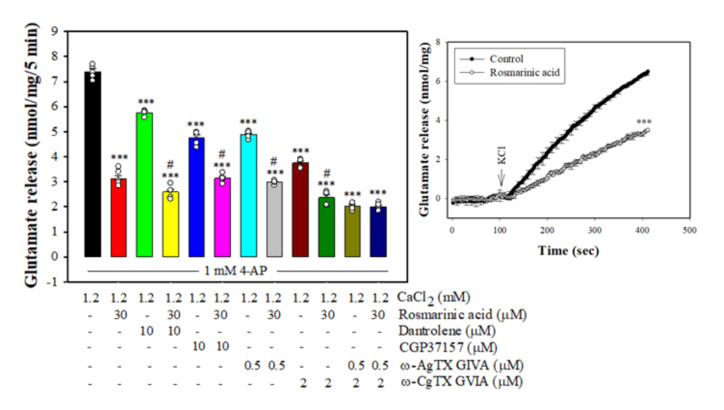
Effect of rosmarinic acid on 4-AP-evoked glutamate release in the presence of N-type Ca^2+^ channel blocker ω-CgTX GVIA, P/Q-type Ca^2+^ channel blocker ω-AgTX IVA, ryanodine receptor inhibitor dantrolene, or mitochondrial Na^+^/Ca^2+^ exchanger inhibitor CGP37157. Inset, the inhibitory effect of rosmarinic acid on the release of glutamate evoked by 15 mM KCl. Rosmarinic acid was added 10 min before the addition of 4-AP, and other drugs were added 10 min before this. Data are mean ± SEM (*n* = 5 per group). *** *p <* 0.001 compared with the control group. ^#^
*p <* 0.001 compared with the dantrolene- or CGP37157-treated group.

**Figure 3 biomolecules-11-01029-f003:**
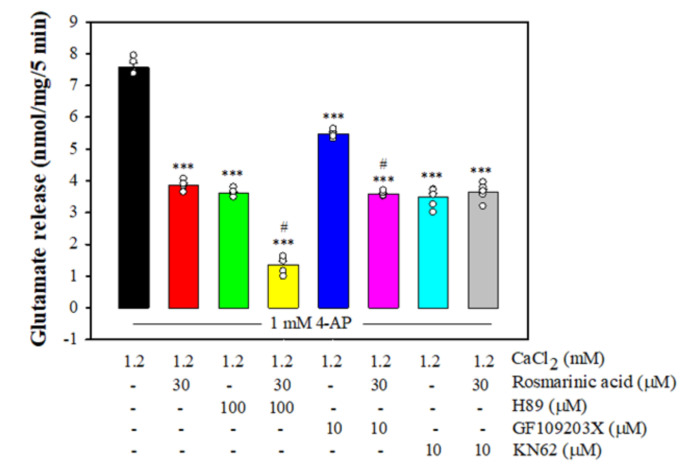
Effect of rosmarinic acid on 4-AP-evoked glutamate release in the presence of the PKA inhibitor H89, PKC inhibitor GF109203X or CaMKII inhibitor KN62. Rosmarinic acid was added 10 min before the addition of 4-AP, and other drugs were added 10 min before this. Data are mean ± SEM (*n* = 5 per group). *** *p* < 0.001 compared with the control group. ^#^
*p* < 0.001 compared with the H89- or GF109203X-treated group.

**Figure 4 biomolecules-11-01029-f004:**
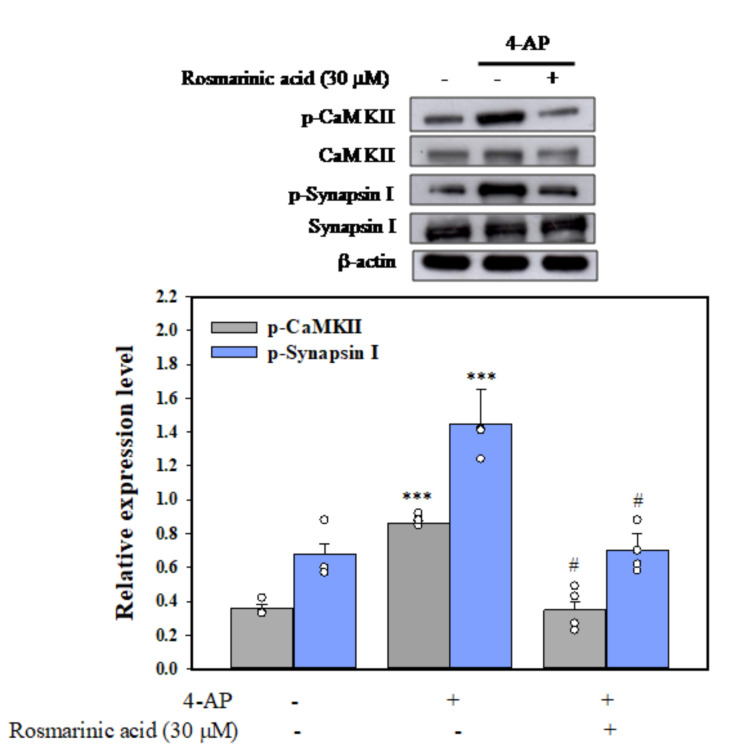
Effect of rosmarinic acid on CaMKII and synapsin I phosphorylation evoked by 4-AP. Rosmarinic acid or KN62 was added 10 min before the addition of 4-AP. Data are mean ± SEM (*n* = 5 per group). *** *p <* 0.001 compared with the control group. ^#^
*p <* 0.001 compared with the 4-AP-treated group.

**Figure 5 biomolecules-11-01029-f005:**
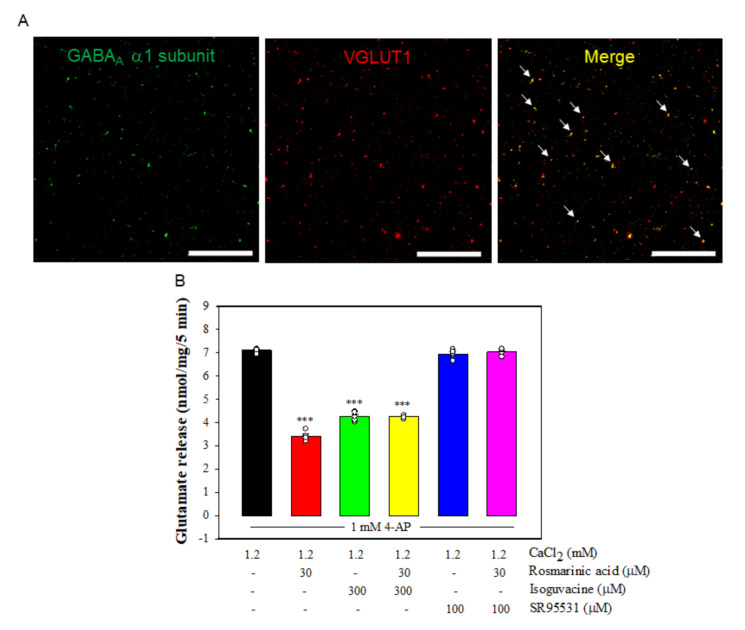
(**A**) GABA_A_ receptorα1 subunits are present in VGLUT1-positive synaptosomal particles isolated from the rat cerebral cortex. Confocal microscopy unveiled a significant colocalization of VGLUT1 (red) and GABA_A_ receptor α1 subunit (green) immunopositivities (yellow, merge, arrowhead). Scale bar: 100 μm. (**B**) Effect of rosmarinic acid on 4-AP-evoked glutamate release in the presence of isoguvacine and SR95531, GABA_A_ receptor agonist and antagonist, respectively. Isoguvacine or SR95531 was added 10 min before the addition of 4-AP. Data are mean ± SEM (*n* = 5 per group). *** *p <* 0.001 compared with the control group.

**Figure 6 biomolecules-11-01029-f006:**
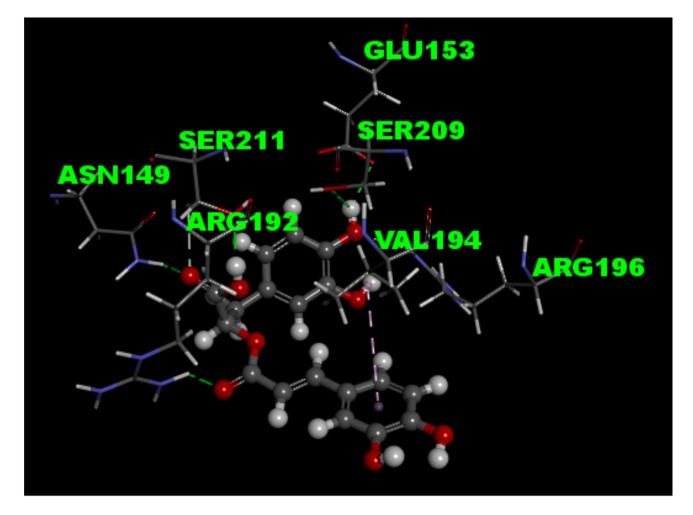
Prediction of the interaction of rosmarinic acid to GABA_A_ receptor protein using molecular docking. Molecular docking modeling of rosmarinic acid with the active site of molecular structure of GABA_A_ receptor protein (PDB ID 4COF) was performed using the Discovery Studio 4.1 software (BIOVIA software Inc., San Diego, CA, USA). Protein–ligand hydrogen-bonding interactions are displayed as green lines.

**Figure 7 biomolecules-11-01029-f007:**
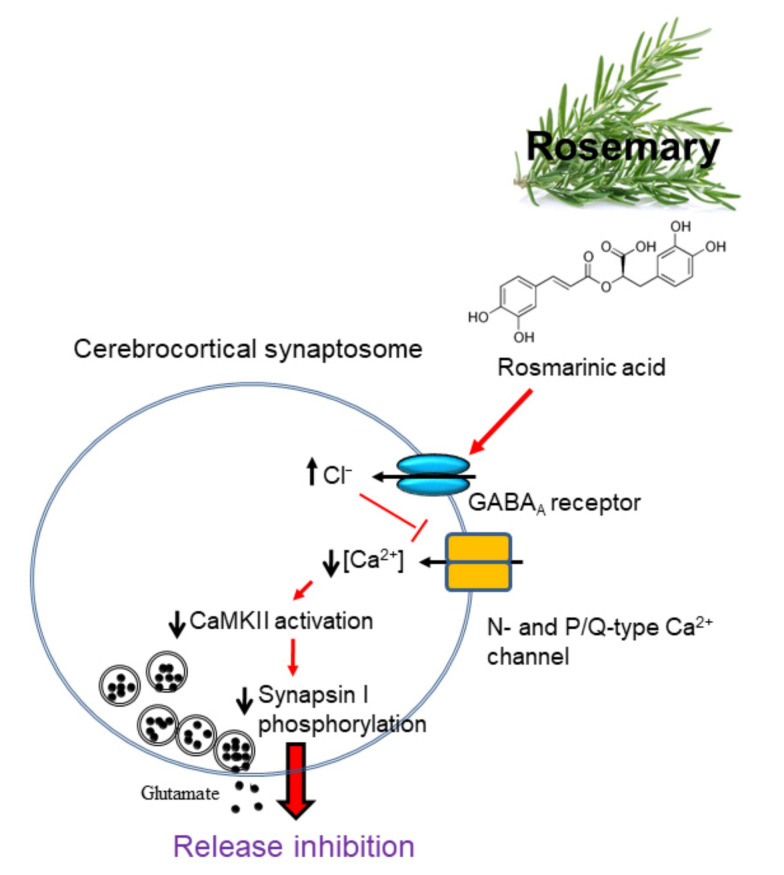
A proposed mechanism underlying the inhibition of glutamate release by rosmarinic acid in rat cerebrocortical synaptosomes.

**Table 1 biomolecules-11-01029-t001:** Effect of rosmarinic acid on cytosolic Ca^2+^ levels and synaptosomal membrane potential in rat cerebrocortical synaptosomes.

	Cytosolic [Ca^2+^] (nM)			DiSC_3_(5) Fluorescence		
	Basal	4-AP (1 mM)	*n*	Basal	4-AP (1 mM)	*n*
Control	148.6±1.1	196.7± 3.1	5	0.6 ± 0.3	24.7 ± 1.6	5
Rosmarinic acid	147.9±2.1	170.7 ± 4.0 ***	5	0.5 ± 0.8	24.3 ± 1.8	5

Data are mean ± SEM (*n* = 5 per group). *** *p <* 0.001 compared with the control group.

## Data Availability

The data presented in this study are available on request from the corresponding author.

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
