# Peer review of "Rosmarinic Acid, a Bioactive Phenolic Compound, Inhibits Glutamate Release from Rat Cerebrocortical Synaptosomes through GABAA Receptor Activation"

_biomolecules, 2021, doi:10.3390/biom11071029_

Round 1

Reviewer 1 Report

This study investigates the biological mechanisms of Rosmarinic acid which is considered to have therapeutic effects. The authours use cerebrocortical synaptosomes extracted from terminals and through a series of experiments conclude that Rosmarinic acid acts through GABAA receptors which lower calcium influx which reduces the CAMKII/synapsin pathway to inhibit glutamate release. They test the model using AP4 to release glutamate from the synaptosomes.

To my understanding the experiments are carried out correctly and the results support the interpretation.

My main questions relate to the GABA receptor part of the study.

Points to address:

As one of the main conclusions in the study is that GABAA receptors are involved in modulation of the glutamate release this needs to be addressed more specifically. GABAA receptors is rather a broad term in their conclusion. In the immunohistochemical methods the Abcam GABAA receptor antibody which was used to show GABAA receptor subunits expressed on their synaptosomes needs to be described and its specificity given particularly in its binding to which particular subunit.

Figure 5A there should be a phase contrast or similar image to show the where the staining is in this field of view is on synaptosomes. Is each dot a synaptosome? Also the resolution of the image should be increased and the degree of colocalization should be indicated with arrows.

In the interpretation of what is the most likely subtype of GABAA receptor that this study has revealed (benzodiazepine , anxiolytic?? or what known subunits are present)  when using the  ligand SR95531. As GABAA receptors are mainly involved in chloride ion channel modulation how do the chloride channels fit into this mix of ion channels?

I am very pleased a diagram of the mechanism related to the study is included.

Author Response

We thank the reviewers for the critical comments and constructive suggestions.

Reviewer 1:

As one of the main conclusions in the study is that GABAA receptors are involved in modulation of the glutamate release this needs to be addressed more specifically. GABAA receptors is rather a broad term in their conclusion. In the immunohistochemical methods the Abcam GABAA receptor antibody which was used to show GABAA receptor subunits expressed on their synaptosomes needs to be described and its specificity given particularly in its binding to which particular subunit.

As suggestion by the reviewer, GABAA receptor a1 subunit is added in the method section (Page 3, line114). In addition, in order to make the statement of the sentence more clear, several sentences are modified (Page 8, Lines 251-256).

Figure 5A there should be a phase contrast or similar image to show the where the staining is in this field of view is on synaptosomes. Is each dot a synaptosome? Also the resolution of the image should be increased and the degree of colocalization should be indicated with arrows.

As suggestion by the reviewer, Fig, 5A is modified. In order to make the statement of the sentence more clear, several sentences are also modified (Page 8, Lines 251-256; Page 9, Lines 271-273).

In the interpretation of what is the most likely subtype of GABAA receptor that this study has revealed (benzodiazepine , anxiolytic?? or what known subunits are present)  when using the  ligand SR95531. As GABAA receptors are mainly involved in chloride ion channel modulation how do the chloride channels fit into this mix of ion channels?

As suggestion by the reviewer, several sentences are added in the discussion section, including GABAA receptors, a large and diverse family of Cl⁻-permeable ion channels, mediate inhibitory neurotransmission in the CNS [49]. GABAA receptors are assembled from seven class of homologous subunits, a, b, g,d, e, p, and q, into heteropentamers. The majority of GABAA receptors are composed of two α1, two β2, and one γ2 subunits (α1β2γ2 receptors), which are primarily located synaptically and are involved in the hypnotic/sedative effects (Page 11, Lines 342-347); Furthermore, our docking data revealed that rosmarinic acid can form hydrogen bond with the amino acid residues (Asn149, Glu153, Arg192, Arg196, Ser209, and Ser 211) of GABAA receptor. Since the amino acid residues Ser204, Tyr205, Arg207, and Ser209 in the loop C region of GABAA protein are the key residues face into GABA binding, and the loop F region (Asp192-Arg197) could interactions with neighboring residues by hydrogen bonding and hydrophobic interaction, which change conformation during receptor activation [54,55]. Rosmarinic acid may be able to modulate the GABAA receptor activation through the interaction between Arg192, Arg196, and Ser209 (Page 11, Lines 352-359). Four references are added in the reference list (Page 14, Lines 501-504, 509-512).

In addition, because the limitation of our experimental technique, the effect of rosmarinic acid on the chloride channels is not able to be done in the present study at present. However, the sentence In the present study, we did not examine the effect of rosmarinic acid on intracellular Cl⁻ influx in cerebrocortical synaptosomes. However, rosmarinic acid has been shown to increase intracellular Cl⁻ influx in primary cultured hypothalamic cells [35] (Page 12, Lines 363-365). Hope you can make allowances for this.

I am very pleased a diagram of the mechanism related to the study is included.

Thank you for your affirmation.

We thank the reviewers for the critical comments and constructive suggestions.

Reviewer 1:

As one of the main conclusions in the study is that GABAA receptors are involved in modulation of the glutamate release this needs to be addressed more specifically. GABAA receptors is rather a broad term in their conclusion. In the immunohistochemical methods the Abcam GABAA receptor antibody which was used to show GABAA receptor subunits expressed on their synaptosomes needs to be described and its specificity given particularly in its binding to which particular subunit.

As suggestion by the reviewer, GABAA receptor a1 subunit is added in the method section (Page 3, line114). In addition, in order to make the statement of the sentence more clear, several sentences are modified (Page 8, Lines 251-256).

Figure 5A there should be a phase contrast or similar image to show the where the staining is in this field of view is on synaptosomes. Is each dot a synaptosome? Also the resolution of the image should be increased and the degree of colocalization should be indicated with arrows.

As suggestion by the reviewer, Fig, 5A is modified. In order to make the statement of the sentence more clear, several sentences are also modified (Page 8, Lines 251-256; Page 9, Lines 271-273).

In the interpretation of what is the most likely subtype of GABAA receptor that this study has revealed (benzodiazepine , anxiolytic?? or what known subunits are present)  when using the  ligand SR95531. As GABAA receptors are mainly involved in chloride ion channel modulation how do the chloride channels fit into this mix of ion channels?

As suggestion by the reviewer, several sentences are added in the discussion section, including GABAA receptors, a large and diverse family of Cl⁻-permeable ion channels, mediate inhibitory neurotransmission in the CNS [49]. GABAA receptors are assembled from seven class of homologous subunits, a, b, g,d, e, p, and q, into heteropentamers. The majority of GABAA receptors are composed of two α1, two β2, and one γ2 subunits (α1β2γ2 receptors), which are primarily located synaptically and are involved in the hypnotic/sedative effects (Page 11, Lines 342-347); Furthermore, our docking data revealed that rosmarinic acid can form hydrogen bond with the amino acid residues (Asn149, Glu153, Arg192, Arg196, Ser209, and Ser 211) of GABAA receptor. Since the amino acid residues Ser204, Tyr205, Arg207, and Ser209 in the loop C region of GABAA protein are the key residues face into GABA binding, and the loop F region (Asp192-Arg197) could interactions with neighboring residues by hydrogen bonding and hydrophobic interaction, which change conformation during receptor activation [54,55]. Rosmarinic acid may be able to modulate the GABAA receptor activation through the interaction between Arg192, Arg196, and Ser209 (Page 11, Lines 352-359). Four references are added in the reference list (Page 14, Lines 501-504, 509-512).

In addition, because the limitation of our experimental technique, the effect of rosmarinic acid on the chloride channels is not able to be done in the present study at present. However, the sentence In the present study, we did not examine the effect of rosmarinic acid on intracellular Cl⁻ influx in cerebrocortical synaptosomes. However, rosmarinic acid has been shown to increase intracellular Cl⁻ influx in primary cultured hypothalamic cells [35] (Page 12, Lines 363-365). Hope you can make allowances for this.

I am very pleased a diagram of the mechanism related to the study is included.

Thank you for your affirmation.

Reviewer 2 Report

The authors describe efforts to determine the mechanism by which rosmarinic acid inhibits gkutamate release from specific rat synaptosomes. In the end, they come to the conclusion, that rosmarinic acid binds to the GABAA receptor and activates it, which leads due to an increased Cl- influx (through the GABAA chanel) to a reduced Ca2+ influx and as the ultimate consequence to glutamate release. The authors support their hypothesis of rosmarinic acid binding to GABAA by a docking study, to the "active site" of the GABAA receptor. Here I ask for more details. Which active site? The binding site of GABA itself or one of the other binding sites (where more than a dozent are known). Since the other binding sites are often allosteric ones, rosmarinic acid could e.g. as positive allosteric modulator, which would go in line with the observations that were made by the authors. There is also previous literature discussing interactions between GABAA receptors and rosmarinic acid (e.g. https://doi.org/10.4062/biomolther.2016.035, https://doi.org/10.3389/fphar.2016.00219). The authors should make a in depth literature search to see whether they can find more support for their hypothesis. Natural, rosmarinic acid could be investigated experimentally to determine whether it is a positive allosteric modulator or not, but such investigations are tricky and would go significantly beyond a simple revision of a manuscript.

To conclude, analysing published data on rosmarinic acid - GABAA interaction should be done and included in a paragraph in the manuscript before it is published. Additionally, the "active site" used for docking should be specified, and of course it should be reevaluated after the aforementioned literature search, whether this can still be considered the proper site for docking.

Author Response

We thank the reviewer for the critical comments and constructive suggestions.

As suggestion by the reviewer, several sentences are added in the discussion section, including GABAA receptors, a large and diverse family of Cl⁻-permeable ion channels, mediate inhibitory neurotransmission in the CNS [49]. GABAA receptors are assembled from seven class of homologous subunits, a, b, g,d, e, p, and q, into heteropentamers. The majority of GABAA receptors are composed of two α1, two β2, and one γ2 subunits (α1β2γ2 receptors), which are primarily located synaptically and are involved in the hypnotic/sedative effects (Page 11, Lines 342-347); Furthermore, our docking data revealed that rosmarinic acid can form hydrogen bond with the amino acid residues (Asn149, Glu153, Arg192, Arg196, Ser209, and Ser 211) of GABAA receptor. Since the amino acid residues Ser204, Tyr205, Arg207, and Ser209 in the loop C region of GABAA protein are the key residues face into GABA binding, and the loop F region (Asp192-Arg197) could interactions with neighboring residues by hydrogen bonding and hydrophobic interaction, which change conformation during receptor activation [54,55]. Rosmarinic acid may be able to modulate the GABAA receptor activation through the interaction between Arg192, Arg196, and Ser209 (Page 11, Lines 352-359). Four references are added in the reference list (Page 14, Lines 501-504, 509-512).

Round 2

Reviewer 1 Report

I am happy to accept these changes. Please change the spelling in the correction below in lines 359-362 from interactions to interact

loop C region of GABAA protein are the key residues face into
 GABA binding, and the loop F region (Asp192-Arg197) could interact with neighbor
ing residues by hydrogen bonding and hydrophobic interaction, which change confor
mation during receptor activation [54,55]. Rosmarinic acid may be able to modulate the
 GABAA receptor activation through the interaction between Arg192, Arg196, and Ser209.

Author Response

We thank the reviewer for the critical comments and constructive suggestions.

As suggestion by the reviewer, the word is changed to interact (Page 11, line 355).